# Measurement of Force and Position Using a Cantilever Beam and Multiple Strain Gauges: Sensing Principles and Design Considerations

**DOI:** 10.3390/s25216561

**Published:** 2025-10-24

**Authors:** Carter T. Noh, Kenneth Smith, Christian L. Shamo, Jordan Porter, Kirsten Steele, Nathan D. Ludlow, Ryan W. Hall, Maeson G. Holst, Alex R. Williams, Douglas D. Cook

**Affiliations:** 1Department of Mechanical Engineering, Brigham Young University, Provo, UT 84602, USA; smith.kenneth@gatech.edu (K.S.); cshamo@byu.edu (C.L.S.); jordanporter0101@yahoo.com (J.P.); kcougar@byu.edu (K.S.); holsma20@byu.edu (M.G.H.); arw86@byu.edu (A.R.W.); 2Robotics Institute, Carnegie Mellon University, Pittsburgh, PA 15289, USA; nludlow@andrew.cmu.edu; 3Department of Mechanical Engineering, University of Utah, Salt Lake City, UT 84112, USA; ryan.w.hall@utah.edu

**Keywords:** force sensor, position sensor, tactile sensing, load cell, strain gauge, end effector, gripper, robotics

## Abstract

**Highlights:**

**What are the main findings?**
Novel sensor concept uses strain gauges on a cantilever beam to simultaneously measure force and position.Sensor is highly linear, with accuracy primarily based on the applied load.

**What is the implication of the main finding?**
The sensor can be designed to fit a wide variety of measurement ranges.The durable, compact design is suitable for rugged field and robotics applications.

**Abstract:**

Simultaneous measurement of force and position often relies on delicate tactile sensing systems that only measure small forces at discrete positions. This study proposes a compact, durable sensor which can provide simultaneous and continuous measurements of force and position using multiple strain gauges mounted on a cantilever beam. When a point force is applied to the cantilever, the strain gauges are used to determine the magnitude of the applied force and its position along the beam. A major advantage of the force-position sensor concept is its compact electronics and durable sensing surface. We designed, tested, and evaluated three different prototypes for the force-position sensor concept. The prototypes achieved an average percent error of 1.71% and were highly linear. We also conducted a thorough analysis of design variables and their effects on performance. The force and position measurement ranges can be adjusted by tuning the material and geometric properties of the beam and the spacing of the strain gauges. The accuracy of force measurements is dependent upon applied load, but insensitive to the location of the applied load. Accuracy of position measurements is also dependent upon applied load and weakly dependent upon position of the applied load.

## 1. Introduction

Force and position sensing are fundamental problems in the field of engineering measurement. Treated alone, these two problems are well defined and have many reliable solutions, but very few sensors can provide continuous (i.e., not discretized) measurements of both quantities simultaneously.

Force measurement is traditionally accomplished in industrial and research applications using load cells. Kamble et al. [1] gives an excellent overview of the various mechanical means of measuring force. The most common approach uses a mechanical structure [2,3,4,5] that converts the applied force to a bending strain. The resulting strain is then measured with one or more strain gauges. Load cells require that the force being measured is applied at a pre-specified location. Inaccurate measurements will result if forces are applied at other positions on the load cell structure or if the load direction is not properly aligned with the load cell. Traditional load cells thus require that the force position be closely controlled.

The problem of measuring an applied force with a variable position has been most thoroughly explored in the field of tactile sensing [6,7]. Tactile sensing is most often accomplished using thin, flexible arrays of individual capacitive [8,9], piezoelectric [10,11], or piezoresistive sensors [12,13]. Positional data is gathered by analyzing which sensors in the array are stimulated. This approach provides only discrete values for position. Simple tactile sensors are made of arrays of small load cells [9,14,15] or single strain gauges placed on deformable surfaces [16,17]. High-fidelity sensors can be made of large, dense sensing arrays with high spatial resolutions [18,19]. A major drawback to tactile sensors is that the delicate sensor arrays can be easily damaged by abrasion and wear. Another limitation is their relatively low force ranges; many tactile sensors are designed to measure contact forces of less than 10 N [20,21]. Measuring larger force ranges often requires sacrificing spatial resolution.

Multiple strain gauges have been used to obtain force and position in previous studies [22,23]. However, the literature surrounding this topic does not provide a comprehensive treatment of this sensing approach. Multiple strain gauges were used in a biomedical application [22], in which three (3) thick-film piezoresistors were used to detect the position and magnitude of a force on a 2-dimensional plate. However, few technical details were provided, and the design principles were not addressed. A similar approach was used in [23] for impact force detection on a plate but required both traditional thin film strain gauges as well as an array of piezo-electric sensors, making for a complex design. In another work, an array of digital optical proximity sensors embedded in an elastomer was used to detect contact forces and position along a beam for use in a robotic gripper [24]. Forces were detected using light diffraction within the elastomer when it was deformed. The elastomer coating, while beneficial for gripping, is susceptible to wear and abrasion, which may affect the sensor’s performance. To prevent damage to the elastomer, the contact forces were limited to 5 N. A double cantilever beam (fixed at both ends) was used in [25]. In that study, strain gauges were mounted at each end to detect the location of a point force. The authors extended this concept to a two-dimensional “smart floor panel” for indoor localization [26], for which the design proves effective. Unfortunately, these studies focused on the measurement of position and did not address the measurement of the applied force. Additionally, the strain gauges at the ends of the beam or panel require more complex wiring and limit the design’s usability in compact or cantilevered applications. Another work used strain gauges at the base and tip of a cantilever beam to measure the beam’s tip deflection in three dimensions due to applied tip forces and moments [27]. However, that study focused on predicting the deflection of the beam rather than the measurement of force and position.

In this paper, we propose a sensor concept that simultaneously measures the magnitude of an applied force and the position of that force along a beam. This approach provides continuous measurement of both force and position using two strain gauges mounted near the base of the cantilever beam. The position of the gauges keeps the electronics of the sensor isolated to one small region, making them compact and easy to protect and leaving most of the cantilever’s length available for measurement. The concept is amenable to applications where durability and space are of concern and where traditional tactile sensors may be too fragile, such as field robotics applications. We also demonstrate that the concept can be used to measure forces in two axes, while other related sensors are generally restricted to a single axis.

The purpose of this paper is to describe how the design parameters of this type of sensor affect sensor performance, including measurement ranges and error. Our goal is not to fully characterize a particular instance of this type of sensor concept, but rather to demonstrate the concept and explore the variables that affect the performance of this sensing approach.

This paper will be organized as follows. In Section 2 (Operating Principles) we derive the force-position sensor concept’s governing equations. Section 3 (Analysis of Performance) provides information on how sensor parameters affect design trade-offs between the ranges of forces and positions that can be measured. Section 4 (Prototypes, Experiments, and Simulations) describes the physical sensor prototypes used in this study, the experiments performed, and simulations used to understand the sensor performance. Section 5 (Results) provides empirical and simulation results in support of the theory developed in Section 2 and Section 3. In Section 6 (Discussion) and Section 7 (Conclusions) we discuss the sensor concept’s limitations and potential applications and provide concluding thoughts.

## 2. Operating Principles

Like many load cells, our sensor concept utilizes resistive foil strain gauges. However, whereas most load cells are designed to measure force at a single position, our concept uses two (or more) strain gauges to simultaneously provide measurement of force and the position of that force, which may be anywhere within a specified physical region.

In its simplest implementation, the sensor concept consists of two strain gauges bonded to a cantilever beam. As shown in Figure 1, these two strain gauges must be located at distinct axial locations. Each strain gauge is incorporated into a separate Wheatstone bridge circuit. If only two strain gauges are used, each will be used in a quarter-bridge configuration. However, additional strain gauges can be mounted at each axial location to form half-bridge or full bridge configurations. For the remainder of this paper, we will use the singular term *strain gauge* to refer to each of the two strain gauge locations.

For an isotropic cantilever beam, the relationship between the bending strain ε and the bending moment M is commonly modeled using the following equation:(1)$$\varepsilon =\frac{My}{EI}$$

Here, E is the Young’s modulus of the beam, I is the beam’s moment of inertia, and y is the distance from the point to the neutral axis. The use of this equation requires that the cantilever beam be homogenous and linearly elastic. It is also assumed that the deflection is static or quasi-static rather than dynamic in nature.

When a perpendicular force F is applied at a distance x from the cantilever’s base, the resulting bending moment experienced by a strain gauge mounted on the beam’s surface a distance $d_{i}$ from the base is given by:(2)$$M_{i}=F\left(x-d_{i}\right)$$

The expression for the strain experienced by the strain gauge is given by:(3)$$\varepsilon_{i}=\frac{y}{EI}F(x-d_{i})$$

As is typical, the strain is transduced to a voltage signal using an amplified Wheatstone bridge circuit. The standard equation relating strain to the output voltage of a Wheatstone bridge for small strains is as follows:(4)$$V_{i}=V_{s}gG\frac{n}{4}\varepsilon_{i}$$

Here $V_{s}$ is the supply voltage, G is the gauge factor of the strain gauge, g is the amplification (gain), and n is a variable relating to the Wheatstone bridge configuration. For quarter, half, and full Wheatstone bridge configurations, n equals 1, 2, and 4, respectively [28]. Combining Equations (3) and (4), we obtain an expression for the voltage in terms of the input force and position.(5)$$V_{i}=\frac{V_{s}Ggny}{4EI}F(x-d_{i})$$

The above equation represents the measured voltage in an ideal scenario. In practice, an imperfectly balanced Wheatstone bridge can be modeled using a modified form of this equation:(6)$$V_{i}=k_{i}F\left(x-d_{i}\right)+c_{i}$$

Here, $c_{i}$ represents bias caused by imbalance in the Wheatstone bridge and $k_{i}$ accounts for all the constant terms from Equation (5):(7)$$k_{i}=\frac{V_{s}Ggny}{4EI}$$

By substituting Equation (2) into Equation (6), we see that voltage is proportional to the applied moment. This is a key concept which guides the analysis in the following section.(8)$$V_{i}=k_{i}M_{i}+c_{i}$$

The sensor concept described in this study makes use of at least two strain gauges mounted on the beam at axial locations $d_{1}$ and $d_{2}$. When a single force F is applied at a distance x from the base, the output voltages from the two strain gauges are:(9)$$V_{1}=k_{1}F\left(x-d_{1}\right)+c_{1}$$(10)$$V_{2}=k_{2}F\left(x-d_{2}\right)+c_{2}$$

Equations (9) and (10) can be solved algebraically for x and F to provide the associated prediction equations:(11)$$x=\frac{k_{2}d_{2}\left(V_{1}-c_{1}\right)-k_{1}d_{1}\left(V_{2}-c_{2}\right)}{k_{2}\left(V_{1}-c_{1}\right)-k_{1}\left(V_{2}-c_{2}\right)}$$(12)$$F=\frac{k_{2}\left(V_{1}-c_{1}\right)-k_{1}\left(V_{2}-c_{2}\right)}{k_{1}k_{2}\left(d_{2}-d_{1}\right)}$$

Figure 2 illustrates the measurement process graphically. Here, the voltage experienced by each strain gauge is plotted as a voltage isoline on a position-force plot, representing all the combinations of x and F that could have caused each strain gauge’s respective voltage curve. The (x,F) point where the two isolines intersect represents the predicted values for force and position. Note that at this point the voltages are not equal. Rather, the voltage is constant along each curve.

## 3. Analysis of Sensor Performance

### 3.1. Sensor Design Limits

In this section, the influence of design parameters on the performance of the force-position sensor concept is addressed.

#### 3.1.1. Maximum and Minimum Moment

The measurement limits of the force-position sensor are defined by the minimum moment the sensor can detect and the maximum moment the device can bear before either plastic deformation or failure occurs in the cantilever beam. These moments are closely related to the maximum and minimum design forces, $F_{max}$ and $F_{min}$.

#### 3.1.2. Maximum Moment and Strain

The maximum moment the sensor can detect is determined by the strain limit of the material. For most metals, plastic deformation occurs above strains of 0.1 [29]. For brittle materials such as ceramics, fracture or failure will define the upper strain limit.

The maximum bending strain will occur at the base of the cantilever (point where cantilever is mounted) when the maximum design force, $F_{max}$, is applied at the tip of the cantilever (furthest point from the base), $x_{max}$:(13)$$\varepsilon_{max}=\frac{F_{max}x_{max}y}{3EI}$$

Once a material is chosen, Equation (13) can be used to assess tradeoffs between $F_{max}$, $x_{max}$, and the cross-sectional design of the beam.

The maximum strain measured by the sensor will be slightly smaller than the strain corresponding to the maximum beam moment because the strain gauge must be located a small distance ($d_{1}$) from the base:(14)$$M_{max}=F_{max}(x_{max}-d_{1})$$

#### 3.1.3. Minimum Moment and Strain

The lower detection limit is determined by the characteristics of the electronic measurement system. The smallest voltage will correspond to a small force applied very close to the strain gauge furthest from the base (strain gauge 2). The sensor’s electronics must be designed so that the smallest desired load, $F_{min}$ applied at the desired start of the measurement region, $x_{min}$ produces a voltage from gauge 2 that is above the latent “noise” of the measurement system. The equation for the minimum voltage that can be measured is found by substituting minimum values into Equation (6):(15)$$V_{min}=k_{2}F_{min}\left(x_{min}-d_{2}\right)+c_{2}$$

The term $F_{min}\left(x_{min}-d_{2}\right)$ is the minimum moment that can be detected, which provides equations for the minimum moment and strain values:(16)$$M_{min}=F_{min}(x_{min}-d_{2})$$(17)$$\varepsilon_{min}=\frac{M_{min}y}{3EI}$$

#### 3.1.4. Operational Limits

Equations (14) and (16) provide two operational limits on the sensor’s performance. Figure 3 illustrates how these limits are related to applied forces and the physical dimensions of the sensor. As shown in this figure, Equation (14) provides a maximum moment isoline and Equation (16) provides a minimum moment isoline. The sensor’s operational range is restricted to the rectangular region in the center of the figure. This region is bounded above and below by maximum and minimum design forces ($F_{max}$ and $F_{min}$) which each intersect with their respective moment isolines.

Figure 3 depicts these concepts. Note that distances in Figure 3 have been normalized by the distance between strain gauges ($\delta =d_{2}-d_{1}$), and forces have been normalized by the minimum design force ($F_{min}$). Additionally, the curves in Figure 3 and Figure 4 do not depict any specific sensor configuration or set of real data but are for illustrative purposes only.

#### 3.1.5. Relationship Between Physical Design and Measurement Limits

The sensor’s cross-section (corresponding to moment of inertia I) and modulus of elasticity (E) should be chosen to facilitate the desired range of forces using Equations (13)–(17). Changes in I and/or E cause vertical shifts in the isolines of Figure 3.

The limits of the operational range also depend upon the length of the measurement region (L), the position of the measurement region relative to the strain gauges (a), and the spacing between strain gauges (δ):(18)$$\delta =d_{2}-d_{1}$$

For clarity, these variables are shown graphically on the sensor diagram in Figure 3.

The range of measurable forces is often a primary design objective. We therefore define the force ratio ($R_{F}$) as the ratio between the maximum and minimum design forces:(19)$$R_{F}=\frac{F_{max}}{F_{min}}$$

The strain ratio has a similar definition and is also important for sensor design. Recall that the maximum strain depends upon the material used and the minimum strain depends upon the electronic measurement system. Because strain and moment are proportional to each other, the strain ratio ($R_{\varepsilon }$) is equivalent to the ratio of moments:(20)$$R_{\varepsilon }=\frac{\varepsilon_{max}}{\varepsilon_{min}}=\frac{M_{max}}{M_{min}}$$

Combining Equations (19) and (20) gives:(21)$$R_{F}=\frac{F_{max}}{F_{min}}=\frac{M_{max}}{M_{min}}\frac{x_{min}-d_{2}}{x_{max}-d_{1}}$$

Next, we substitute the definitions of a, L, and δ to change the design variables from $d_{1}$ and $d_{2}$ to the locally referenced a and δ. The force ratio is found to depend upon the design parameters in Figure 3 as follows:(22)$$R_{F}=R_{\varepsilon }\frac{a}{a+L+\delta }$$

Equation (22) reveals useful insights regarding the relationships between design variables and sensor performance. First, the variable $d_{1}$ has no effect on the force ratio when a, L, and δ are held equal. That is, if the cantilever is increased in length and the strain gauges and measurement region are shifted forward accordingly, the range of measurable forces is unchanged. However, increasing $d_{1}$ in this way will significantly affect the deflection of the beam. If beam deflection is to be minimized, $d_{1}$ should be minimized by placing it as close to the cantilever’s base as possible. If beam compliance is a desired attribute, $d_{1}$ could be adjusted accordingly.

Second, the distances a, L, and δ can each be used to adjust the force ratio. The force ratio is most effectively increased by decreasing L and/or δ. Figure 4 provides an illustration showing two contrasting sensor designs. Design 1 has a high range of measurable forces but a small range of measurable positions. In contrast, Design 2 has a broad positional measurement region but lower range of measurable forces. Note that the vertical axis in Figure 4 is linear, whereas this axis is depicted with a logarithmic scale in Figure 3.

### 3.2. Voltage Disparity and Sensor Accuracy

Ideally, the constants $k_{i}$ will be identical for each strain gauge (see Equation (7)). If we assume that $k_{1}=k_{2}=k$ and that all Wheatstone bridges are perfectly balanced ($c_{1}=c_{2}=0$), Equations (11) and (12) reduce to the following:(23)$$x=\delta \left(\frac{V_{1}}{∆V}\right)-d_{1}=\delta \left(\frac{V_{2}}{∆V}\right)-d_{2}$$(24)$$F=\frac{∆V}{k\delta }$$
where ∆V is the voltage disparity ($V_{1}-V_{2}$). These equations show that F is directly proportional to ∆V and x is proportional to the ratio ($V_{1}/∆V$). In other words, the sensor largely operates upon the voltage difference between strain gauges. Solving Equation (24) for ∆V shows that the voltage disparity is dependent upon k, F, and δ.

As the distance between strain gauges increases, the two voltage isolines in Figure 2 become more distinct and the angle of intersection increases. This is illustrated in the left-hand panel of Figure 5. The solution uncertainty is inversely related to the angle of intersection between the two voltage isolines. When the angle of intersection is small, the uncertainty region will be large, and vice versa. Thus, a larger distance between strain gauges will result in less uncertainty in the solution. This effect is depicted in the center and right-hand panels of Figure 5.

Substituting the terms that constitute k from Equation (7) and collecting terms allows us to assess how various design choices affect the voltage disparity:(25)$$∆V=\frac{1}{4}V_{s}gGn\delta \frac{y}{EI}F$$

Equation (25) shows that the voltage disparity is influenced by electronics design factors, strain gauge configuration, the beam’s geometry and material properties, and the applied force. The electronic factors include the excitation voltage ($V_{s}$) and the signal amplification or gain (g). The strain gauge factors include gauge factor (G), the number of strain gauges used in each Wheatstone bridge (n), and the spacing between strain gauges (δ). The beam geometry affects the voltage disparity inversely through the beam’s section modulus (S=y/I). Finally, the elastic modulus of the beam (E) also has an inverse effect on voltage disparity. These proportional and inverse relationships should be considered when designing for a particular application.

### 3.3. Calibration and Prediction

The operational range shown in the above Figures is important in the calibration process. Although the $k_{i}$ values for a given implementation of this sensor could be obtained from individual measurements of all the constants in Equation (7), this approach is prone to stack-up error in the assessment of the individual constants. A better approach is to obtain $k_{i}$ and $d_{i}$ values through empirical calibration of the sensor.

The process described below provides an efficient method for obtaining reliable estimates of $k_{i}$, $d_{i}$ and $c_{i}$ values through calibration rather than through direct measurement of the constituent constants. It uses simple point loading to calibrate the sensor, which is a cheap, easy and effective calibration method. Other calibration techniques do exist for cantilever force sensors such as scanning along the cantilever with a known force [30,31,32], but this requires accurate measurement of tip deflection and application of constant force under movement, which can be difficult or expensive. As will be shown, accurate calibration can be obtained using the following technique with very few point measurements, making it suitable for a broad range of applications.

We start by rearranging Equations (9) and (10) as follows:(26)$$V_{i}=k_{i}Fx-k_{i}d_{i}F+c_{i}$$

Using a change in variables $Fx\to z_{1}$ and $F\to z_{2}$, the equation above is converted into the standard form for multiple linear regression:(27)$$V_{i}=\beta_{0,i}+\beta_{1,i}z_{1}+\beta_{2,i}z_{2}$$

With three unknown constants in Equation (27), a minimum of three measurements is required to perform calibration for each strain gauge. Calibration measurements can be obtained by placing the sensor in a horizontal position then hanging calibration masses vertically at known positions and recording the resulting voltages from the strain gauges.

A general principle of calibration is that calibration points should span the entire measurement region. This approach maximizes regression accuracy and prevents errors associated with extrapolation. To fully span the measurement region, the recommended minimum calibration scheme is to use the four corners of the operational measurement region. Additional points can be used if a more accurate calibration is desired. Figure 6 shows the minimum recommended calibration points as black dots with supplementary calibration points as white dots.

Once calibration data has been collected, least squares regression is used to solve for the unknown coefficients in Equation (27). The individual calibration constants are then obtained as:(28)$$c_{i}=\beta_{0,i}$$(29)$$k_{i}=\beta_{1,i}$$(30)$$d_{i}=\beta_{2,i}/\beta_{1,i}$$

One set of calibration tests is sufficient to calibrate both strain gauges if voltages from both strain gauges are recorded simultaneously. Once the values of $k_{i}$, $d_{i}$ and $c_{i}$ have been obtained for each strain gauge, measurements of applied forces and positions are possible based upon voltages from the two strain gauges using Equations (11) and (12).

## 4. Prototypes, Experiments, and Simulations

### 4.1. Prototype Construction

To validate the sensor concept and demonstrate the effects of design variables, three different prototype designs were constructed, as shown in Figure 7. Detailed dimensions for each prototype are included in Table 1.

For each prototype, DAOKI BF120-3AA 120 Ω metal foil strain gauges were bonded by hand to the cantilever beam at the specified locations using cyanoacrylate glue. Wheatstone bridges consisting of strain gauges and 120 Ω resistors were soldered by hand, or for Prototype C, embedded in a printed circuit board.

Prototype A was constructed using a rectangular aluminum bar. Strain gauges were mounted on the top and bottom of the bar, left and right of the centerline, for a total of 8 strain gauges wired into two full bridges.

Prototype B consisted of a square bar of aluminum with strain gauges mounted on the top and bottom of the beam in half bridges at three locations. At each strain gauge location, a 0.95 cm diameter hole was drilled through the beam. These holes act to amplify the strain at the location of the strain gauges. The use of such stress concentrators is common in load cell designs.

Prototype C was built using a hollow, circular aluminum rod with two sets of strain gauges. The purpose in doing so was to create a bi-axial sensor that could detect position along with two components of force. Diagrams and a photo of Prototype C are provided in Figure 8.

### 4.2. Prototype Calibration and Measurement

#### 4.2.1. Data Collection

A data set was generated for each prototype by performing a series of measurements using known forces applied at known positions. Forces were applied to each prototype using a set of calibrated masses. The beam of each prototype was oriented horizontally, and masses were hung vertically from the prototype at specified positions along the beam’s length, as shown in Figure 9. To control the location at which loads were applied, thin grooves were machined into the surface of each prototype using a high-precision CNC (computer numerical control) machine. Weights were hung from a very thin string placed within these grooves to ensure precise and repeatable positioning of the known loads. The masses and locations for each prototype are provided in Table 2.

To assess the validity of biaxial sensing capability of Prototype C, weights were hung with the device rotated about the sensing axis at angles of 0, 15, 30, 45, 60, 75, and 90 degrees.

#### 4.2.2. Calibration Data

Two subsets of the collected data were used to perform calibration. In the first subset, four force/position data points were reserved for this purpose. These four points corresponded to the 2 × 2 grid of highest and lowest forces and positions. In the second subset, a 3 × 3 calibration grid was used for a total of nine calibration data points. For both subsets, calibration parameters k, d, and c were obtained for each prototype using the calibration process described above.

#### 4.2.3. Measurement and Validation Data

The remaining data (all data not used for calibration) was used to assess the predictive accuracy of each prototype. In the validation stage, recorded voltages were used in conjunction with Equations (11) and (12) to measure the corresponding applied force and its position. Measurements were compared to the true applied force and position to assess the accuracy of each prototype.

Prototype A had a single pair of strain gauge bridges, resulting in one set of measurements for force and position. Prototype B had three sets of strain gauges, resulting in three sets of measurements (one for each combination of two strain gauges). However, only the set using the outer (first and third) gauges was used in this experiment. For prototype C, two pairs of strain gauge bridges were used in perpendicular axes, resulting in two measurements of force and position in two different axes. The two force measurements provide two components of the applied force, which we use to calculate the magnitude F and angle θ that the force is applied in the plane perpendicular to the axis of the cantilever beam. The two position measurements are combined using a weighted average based on this predicted angle to determine the position x of the applied force.

#### 4.2.4. Influence of Strain Gauge Spacing

A set of experiments were conducted to demonstrate the effect of strain gauge spacing on measurement accuracy. In these experiments, all three sets of strain gauges on Prototype B were utilized. This allowed the sensor to be used in two configurations, with strain gauge spacing of 4 cm and 8 cm.

Tests were performed with 11 load values (20, 50, 70, 100, 200, 500, 1000, 1500, 2000, 2500, and 3000 g). Each mass was hung at 7 position values (15, 20, 25, 30, 35, 40, and 45 cm) for a total of 77 unique loading configurations. Measured forces and positions from each strain gauge spacing were compared with known values to obtain relative error as functions of strain gauge spacing, applied load, and position.

This experiment was repeated 5 times to obtain repeatability estimates as functions of applied load and position.

### 4.3. Simulations for Assessing Variation in Performance Across the Operational Range

To gain a more comprehensive understanding of how sensor performance varies across the operational range, a Monte Carlo analysis was performed. This analysis simulated the influence of voltage and calibration uncertainty on sensor accuracy. The simulation was structured to match the physical and electrical characteristics of Prototype B described above. Equations (9) and (10) were modified to include random variable (ϵ) that modeled uncertainties due to voltage measurement and model fit:(31)$$V_{1}=k_{1}F\left(x-d_{1}\right)+c_{1}+\epsilon_{1}$$(32)$$V_{2}=k_{2}F\left(x-d_{2}\right)+c_{2}+\epsilon_{2}$$

The total voltage uncertainty (ϵ) from both factors was defined using a mean of 0 and a standard deviation of 1.3 mV (value based on actual measurements). The measurement range was specified by a grid defined by 7 mass loads ranging from 200 g to 2000 g and 7 measurement positions ranging from 15 to 45 cm. Finally, as with Prototype B above, two values of strain gauge spacing were used: δ = 4 cm and δ = 8 cm. These values match the experimental values used to calibrate and validate Prototype B in Section 4.

For each point in the grid, 100,000 possible voltage values were simulated using Equations (31) and (32). These voltage values were then used to solve for the corresponding load and position values using Equations (11) and (12). The relative errors were then computed at each point in the operational range.

## 5. Results

### 5.1. Illustration of Sensor Data

To illustrate the type of data obtained with this sensor, we start by showing results of a simple experiment in which a fixed load (1000 g mass) was placed sequentially at positions 15, 20, 25, 30, 35, 40, and 45 cm on Prototype B. As seen in Figure 10 below, both the mass load and positions were correctly identified.

Similarly, a mass of 100 g was hung at the 30 cm position on the same sensor. Masses were added sequentially in increments of 100 g up to 500 g. The results are provided in Figure 11. The process of adding mass caused transient disruptions of the signal, but once the transients decayed, the correct loadings were registered. The position was registered at ~30 cm throughout this experiment because the mass load was never entirely removed from the sensor. The right-hand panel of Figure 11 shows that the noise levels in the measured position decreased through the experiment as the load was increased.

### 5.2. Calibration Results

Calibration accuracy was assessed by computing the R^2^ value for each prototype’s calibration. For each prototype, calibration was first conducted using a 2 × 2 grid of four calibration points spanning the sensor’s force and distance measurement regions, then using a 3 × 3 grid of 9 calibration points. Overall, the calibration process exhibited very high R^2^ values with a median value of 0.9999 and a minimum value of 0.99917, showing that the prototypes exhibit highly linear behavior. Table 3 reports the individual R^2^ values.

Both 4-point and 9-point calibration methods produced regression fits with very high R^2^ values. This result indicates that the point-loading method provides a very accurate calibration. Additionally, no significant difference was seen between the 4- and 9-point calibrations. Thus, for most applications, 4-point calibration is likely sufficient.

### 5.3. Measurement Predictions

Measurement linearity was assessed by plotting the predicted values of F and x (and θ for Prototype C) against the corresponding true values. For a perfect sensor, all measurements would lie on a line with slope 1, indicating that the measurement corresponds perfectly with the true applied value. To evaluate the linearity of the prototypes, the R^2^ value corresponding to the regression of the measurement data points with this 1:1 line was calculated for each measurement. These results are shown in Figure 12.

The best-fit lines each have a very high R^2^ correlation, showing that all the sensor prototypes are highly linear. This is to be expected, as the outputs of the measurement Equations (11) and (12) are both linear in relation to the measured voltages.

Table 4 gives the mean error for each prototype and each measured value. All prototypes have an absolute error below 2.75%, with a total mean error across all prototypes of 1.713%.

### 5.4. Influence of Strain Gauge Spacing on Measurement Accuracy

In agreement with theory, the spacing of the strain gauges was inversely related to sensor accuracy. When the spacing of strain gauges was increased, error was reduced at all measured percentile locations, as shown in Figure 13.

### 5.5. Performance as a Function of Applied Load and Position

Sensor performance was found to be more sensitive to applied load than the position of the applied load. Using the more accurate δ = 8 cm configuration on Prototype B, relative errors in measured force and measured position were plotted as functions of applied load. The results are shown in Figure 14. As seen in this figure, percentage errors in both force and position were highest for low applied loads with errors decreasing as loads increased. Errors for both force and position were less than 1% for loads above 1000 g.

The sensor exhibited no significant sensitivity to the position of the applied load. The relationship between position and measurement error for Prototype B with a strain gauge spacing of δ = 8 cm and loads above 200 g is shown in Figure 15. For this configuration, virtually all measurement errors were below 4% with the majority below 2%.

### 5.6. Simulation Results

The entire Monte Carlo simulation process consisted of 4.9 million simulations of the sensor’s performance. These simulations provided a more comprehensive visualization of the patterns described in the preceding sections. Contour plots were created to visualize the way errors in predicted load and position are influenced by location within the operational range and by strain gauge spacing.

First, simulation results indicate that errors in predicted force are highly dependent upon load, moderately dependent upon strain gauge spacing, and insensitive to the position of the applied load. Contour plots showing error in predicted force are shown in Figure 16.

Position predictions were also highly dependent upon applied load and moderately dependent upon strain gauge spacing. However, unlike force predictions, position predictions were also weakly dependent upon position. Higher positions exhibited higher errors than lower positions. Contour plots of errors in predicted position are provided in Figure 17.

These effects can be explained mathematically by looking at the definition for the error in force and position:(33)$$e_{F}=1-\frac{F_{m}}{F_{a}}$$(34)$$e_{x}=1-\frac{x_{m}}{x_{a}}$$

Here $e_{F}$ and $e_{x}$ are the errors in force and position, $F_{m}$ and $x_{m}$ are the true applied force and position, and $F_{a}$ and $x_{a}$ are the measured force and position. By plugging in (23) and (24) for $F_{m}$ and $x_{m}$ and adding noise as in (31) and (32), we can derive the following relationships:(35)$$e_{F}=\frac{\epsilon_{2}-\epsilon_{1}}{{k\delta F}_{a}}$$(36)$$e_{x}=2\frac{d_{1}}{x_{a}}-\frac{\epsilon_{1}}{{kF}_{a}}$$

Equation (35) tells us that the error in force is inversely proportional to the applied load—as the load increases, the error decreases. We also note that the error in force is not dependent on the applied position at all. These corroborate the simulation results in Figure 16. Similarly, Equation (36) matches the results in Figure 17. As the applied position becomes large relative to $d_{1}$, its effect on the positional error decreases. The inverse dependence on applied load is seen as well.

## 6. Discussion

### 6.1. Synthesizing Results

The experimental results demonstrate that this type of sensor is effective in the simultaneous measurement of force and position. In the first experiment with all three prototypes, the large R^2^ values for both calibration and measurement demonstrate the device’s highly linear behavior, as is expected from its underlying measurement equations. It also achieves good predictive accuracy, with error below 2.75%.

Figure 16 and Figure 17 provide an excellent “big picture” overview of how errors are distributed across the operational range. As seen in these figures, there is a broad range of forces and positions with high accuracy (low errors). Figure 14 and Figure 15 reinforce this concept with empirical evidence.

Although errors are generally low, they are not distributed uniformly. Errors in predicted force were found to be dependent upon load, but insensitive to position (Figure 14, Figure 15 and Figure 16). In contrast, errors in predicted position were dependent upon load and weakly dependent upon position (Figure 14 and Figure 17). Note that the relatively minor effect of position on errors in position predictions is not apparent in Figure 15 because it is obscured by the much larger influence of load on these errors. Finally, the results also consistently indicate that higher strain gauge spacing will provide higher accuracy than a lower spacing between strain gauges.

One critical aspect of the sensor’s operation is its ability to measure both force and position in continuous space, rather than at discrete values. Therefore, the “resolution” of the spatial and force measurements is limited only by the electronic components that read the voltages form the strain gauges. However, the sensor’s accuracy, as measured by percent error from the true applied values, is dependent on gauge spacing and applied load and position, and cannot be made arbitrarily small. For this reason, we do not characterize the sensor’s “resolution” but instead provide analysis of its accuracy.

### 6.2. Potential Applications

The sensor concept presented in this paper is capable of simultaneous, continuous measurement of both force and position, and is adaptable to a variety of measurement ranges. Because of its simple structure, it is easily designed for durability and reliability in harsh or rugged conditions. The sensor is also compact, requiring only a single cantilever and two strain gauges.

This sensor could be useful in industrial automation settings, where the weight and position of an object might be important information for a process and where sensors need to work reliably under constant, possibly rough use. This sensor concept may be particularly beneficial in situations where the position of objects is difficult to control precisely (for example, in agricultural processing and agricultural robotics). The sensor can also be integrated into the structure of the automation process, making it more compact and simpler than other force-position sensor combinations, which may require complex structures to distribute forces to a load cell, extra space for another sensor to detect position, and additional processing of the raw signals to obtain accurate measurements.

Another potential application is in robotic gripping. Like the gripper in [24], two of these sensors could be mounted with the cantilevers in parallel on linear actuators. The actuators bring the two cantilevers together to close on either side of an object. When the object is touched by the cantilevers, the force and location measurements from each sensor could be used to determine the location of the object on the gripper and the contact forces. These are important factors for effective robotic gripping that can improve gripping effectiveness. The continuous sensing of force and position has a distinct advantage over other gripping sensors that provide only discrete position values or limited resolution.

### 6.3. Assumptions and Limitations

In this section we briefly outline the assumptions and limitations that should be considered in interpreting this work.

#### 6.3.1. Linearity Assumptions

The derivation of the prediction Equations (10) and (11) in Section 2 relies on several assumptions. Each limits the ways this sensor concept can be used, and breaking these assumptions may affect the performance of the sensor. The following paragraphs cover these assumptions.

The use of Equation (1) requires three assumptions. First, the cantilever beam must be constructed of a material that is linearly elastic. This means the applied loads must remain small enough to prevent plastic deformation or failure. Second, the model formulation assumes that the beam is homogeneous in shape and material properties along the length of the beam. Third, the system must be static or quasi-static in nature. This limits the device to uses where dynamic effects can be neglected.

The use of Equation (2) requires that the deflection of the cantilever beam remains small relative to its length. This assumption is needed because under large deflections the moment becomes dependent on the amount of deflection, thus invalidating Equation (2). This may, however, be a forgiving constraint; our testing on Prototype A, which used a thin, wide beam, resulted in a max tip deflection of 26.5% of the beam length, yet the calibrated device still yielded remarkably linear measurements.

The use of Equation (4) requires that the change in resistance of the strain gauges in the Wheatstone bridge remain small. This is due to the weakly nonlinear nature of the Wheatstone bridge. However, the strains experienced in most strain gauge applications are not large enough to have significant nonlinear effects, and linearity in the Wheatstone bridge is a standard assumption in strain gauge measurements [33].

Equation (2) also assumes that the applied force acts at a single point on the beam. Measurement of multiple point loads or applied moments is not possible with this sensor concept as it is currently formulated. However, the sensor can measure the equivalent force and location corresponding to a non-point load, if the load is distributed over a relatively small region of the beam. The ability of the sensor to identify the equivalent force and location of loads distributed over large regions of the beam was not investigated in this study.

Finally, Equation (2) assumes that the load is applied perpendicular to the beam. Loads that are not perpendicular to the beam will have a perpendicular component and an axial component. If temperature-compensating Wheatstone bridges are used, the axial component of the load will have no effect on the resulting voltage signals. The sensor will then register only the perpendicular component of the applied load (but not the full magnitude of the applied load). Note that the inferences in this paragraph are theoretical in nature and were not investigated as part of this study.

#### 6.3.2. Limitations in Error Analysis

The error analysis presented in this work was conducted on only one prototype. It was not intended to be a thorough sensor characterization, but rather an analysis of the principles that contribute to error. The results in Section 5.5 and Section 5.6 should be verified for specific sensors.

In addition, the error analysis only considered errors in measurement linearity. It did not include investigations into the effects of creep, hysteresis, repeatability, or other effects common for force sensors. Future work can address these types of errors in the application of the sensor concept to specific problems.

## 7. Conclusions

Measurement of force and position are common problems in engineering, but there are few if any sensor solutions that can provide *continuous* measurements of both quantities simultaneously. We have developed, modeled, analyzed, and validated a sensor concept that is capable of simultaneous and continuous measurement of both force and position. The force-position sensor concept consists of at least two strain gauges mounted at two positions along a cantilever beam. Three separate prototypes were constructed, calibrated, and validated. Across all three prototypes, the sensor proved to be highly linear with a predictive R^2^ value of over 0.98 for force and position measurements and achieved an average percent error of 1.71% across all measurements. The sensor’s accuracy in both position and force measurements was shown to be dependent on the applied load, and measurements of position are weakly dependent on the location of the applied load. The sensor can be designed for a wide variety of force and position measurement ranges. Having no moving parts and compact wiring, the sensor can be designed to be space efficient, rugged, and relatively easy to manufacture.

This force-position sensor is only an initial concept and has room for improvement. Future work could focus on optimizing the device for specific applications, such as robotics or industrial automation. Each of these applications would require further design and proper validation to successfully adapt the sensor concept into a usable, reliable measurement device. This includes designing the cantilever beam material, size and profile to measure the desired strains while minimizing deflection. Validation of proper manufacturing of the cantilever and strain gauge application is also necessary to ensure accuracy and long-term reliability.

Along with application-specific design, additional work might also focus on expanding the usability of the sensor concept by addressing its limitations. Measurement of axial forces, multi-component forces, and force distributions instead of only perpendicular point forces are open problems. Similarly, improvements could extend the sensor concept from the measurement of forces along a single linear axis to planes and 3-dimensional surfaces. Work could also focus on extending the sensor concept to allow the use of anisotropic, nonlinearly elastic materials for the cantilever or to allow large beam deflections.

## Figures and Tables

**Figure 1 sensors-25-06561-f001:**
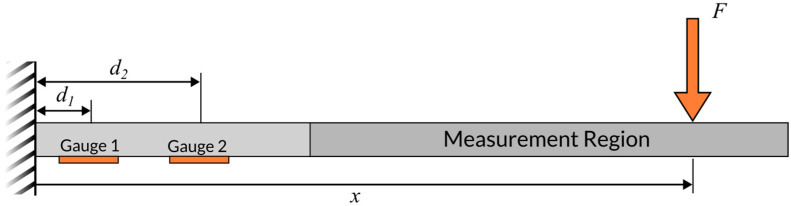
Generic sensor architecture: a cantilever beam with strain gauges bonded at two axial locations. Key variables include the strain gauge positions, force position and force magnitude.

**Figure 2 sensors-25-06561-f002:**
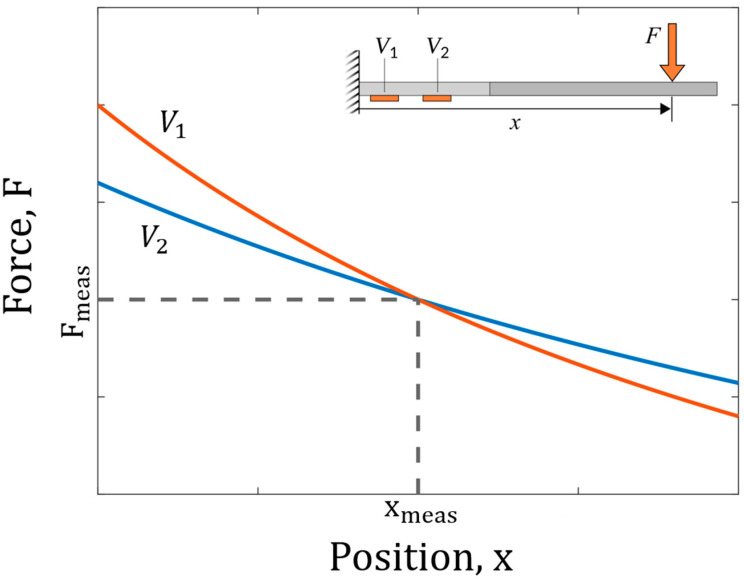
Voltage isolines for two strain gauges. The intersection of the voltage lines represents the actual applied force, F and position where the force is applied, x.

**Figure 3 sensors-25-06561-f003:**
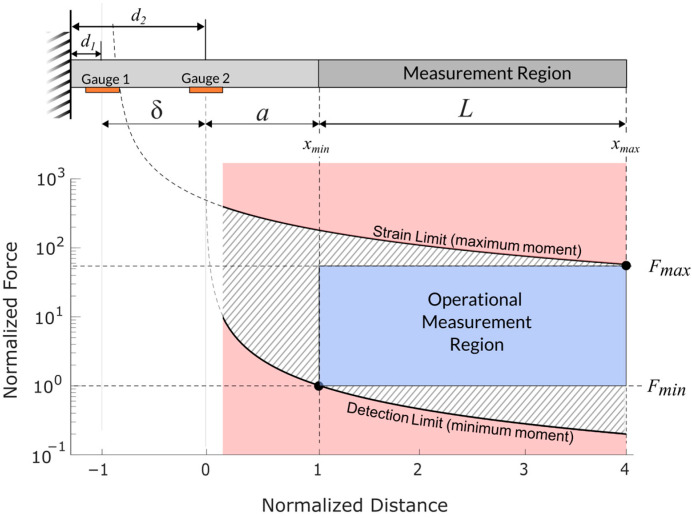
Chart illustrating cantilever dimensions, strain limits, and operational measurement region. The physical layout of the sensor is shown above the chart, with strain gauges (gold), physical measurement region (darker gray, L), spacing between strain gauges (δ), and distance between the second strain gauge and the start of the measurement region (a). Note that while a and δ appear similar here, they need not be in application. Minimum and maximum design forces are shown on the right-hand side. Maximum and minimum moment isolines determine the upper and lower corners of the operational measurement region. The horizontal axis has been adjusted so that 0 is at strain gauge 2 and the units are in terms of the distance between strain gauges. The vertical axis has been normalized by the minimum design force. Diagonal hatching represents measurements which may be possible but are outside of the operational measurement region.

**Figure 4 sensors-25-06561-f004:**
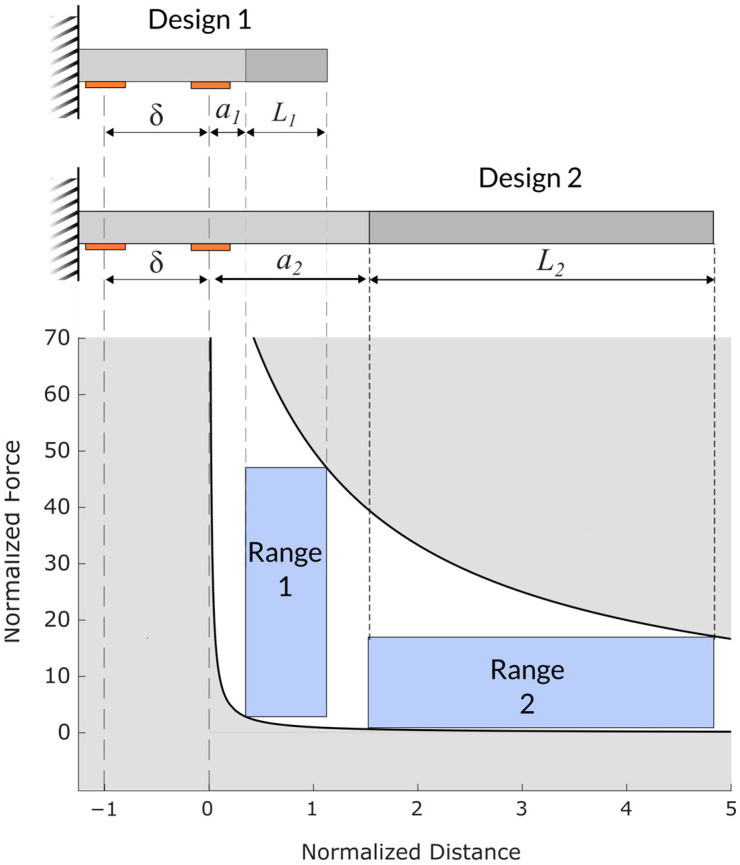
Diagrams showing two possible sensor designs and their associated measurement regions. The variables a, L, and δ from Equation (21) are shown for both designs. (**Top**) designs shown and variables defined. (**Bottom**) The operational measurement region for the two sensor designs.

**Figure 5 sensors-25-06561-f005:**
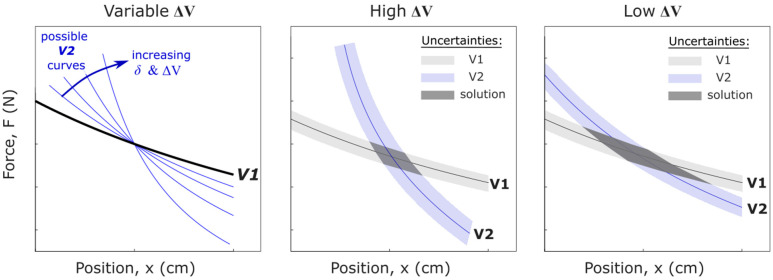
(**Left**) As the spacing between strain gauges (δ) increases, so too does the difference in voltages and the angle between the intersection of the $V_{1}$ and $V_{2}$ isolines. (**Center**) The solution uncertainty region is the overlap between voltage uncertainty curves. Solution uncertainty is decreased with a high δ (high ∆V). (**Right**) Solution uncertainty is increased with a low δ (low ∆V).

**Figure 6 sensors-25-06561-f006:**
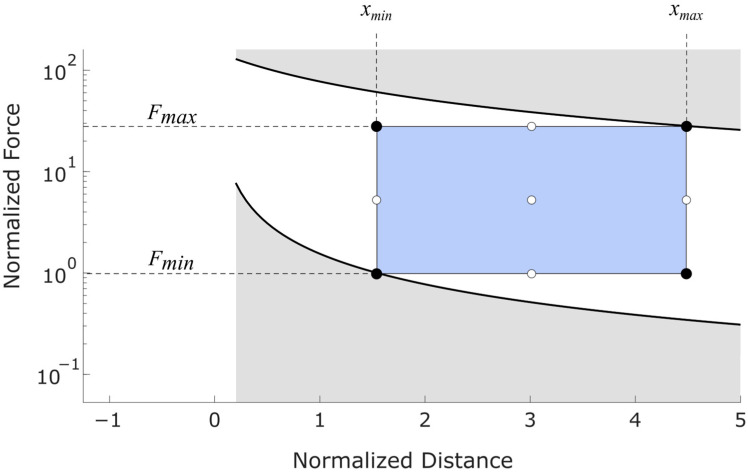
Calibration points relative to the operational measurement region (blue rectangle) Normalized axes are the same as Figure 3.

**Figure 7 sensors-25-06561-f007:**
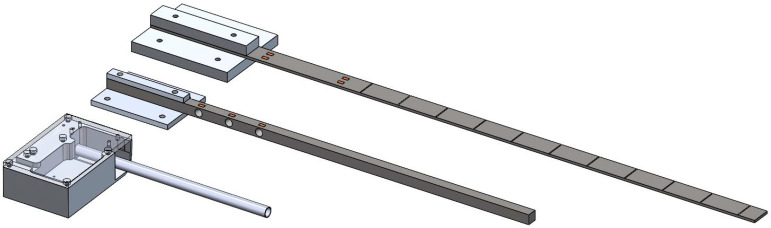
CAD models of Prototypes A (**top right**), B (**middle**), and C (**bottom left**). Strain gauges are shown in orange.

**Figure 8 sensors-25-06561-f008:**
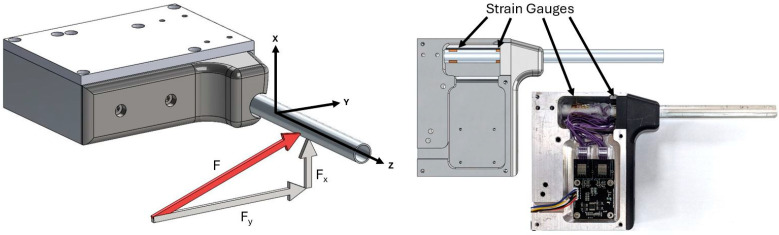
(**Left**) CAD for Prototype C showing components of the applied force. (**Right**) Prototype C CAD compared with physical prototype, with location of strain gauges highlighted.

**Figure 9 sensors-25-06561-f009:**
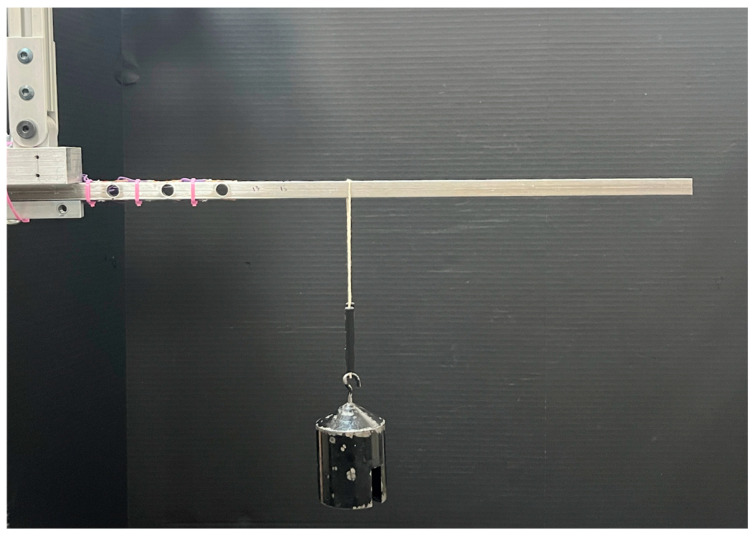
Method for calibration and measurement experiments. Prototype B shown.

**Figure 10 sensors-25-06561-f010:**
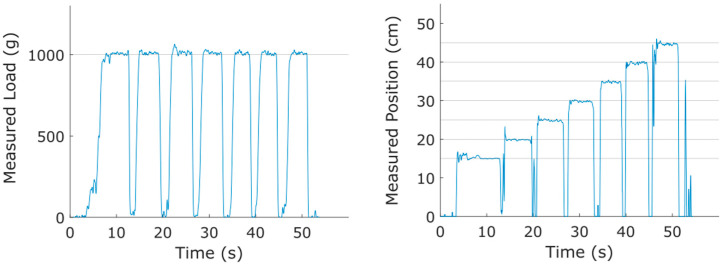
Results of fixed mass (1000 g), variable position experiment. Positions were 15 through 45 cm in 5 cm increments.

**Figure 11 sensors-25-06561-f011:**
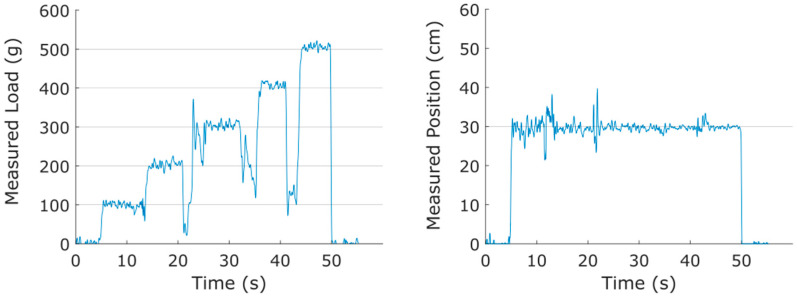
Results of fixed position, variable mass experiment. Masses of 100, 200, 300, 400, and 500 g were hung at the 30 cm position on the sensor.

**Figure 12 sensors-25-06561-f012:**
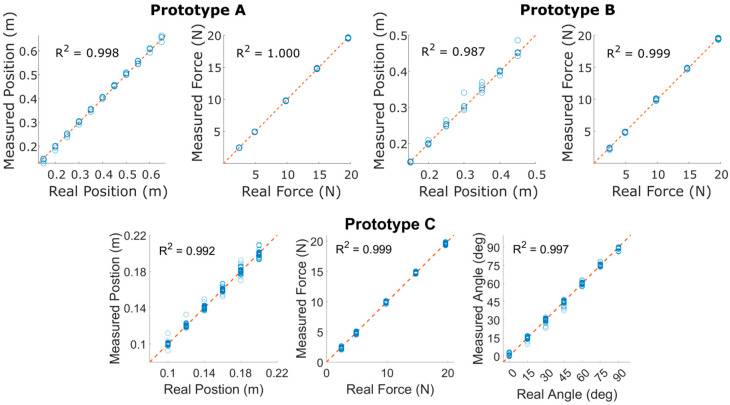
Position, force, and angle measurements vs. actual values for prototypes A, B, and C. The red line indicates a 1:1 line between predicted and true. Measurements closer to this line are more accurate. Each plot displays the R^2^ value for the fit of the data to this 1:1 line. Prototype A, B, and C have sample sizes of n = 53, 35, and 210, respectively.

**Figure 13 sensors-25-06561-f013:**
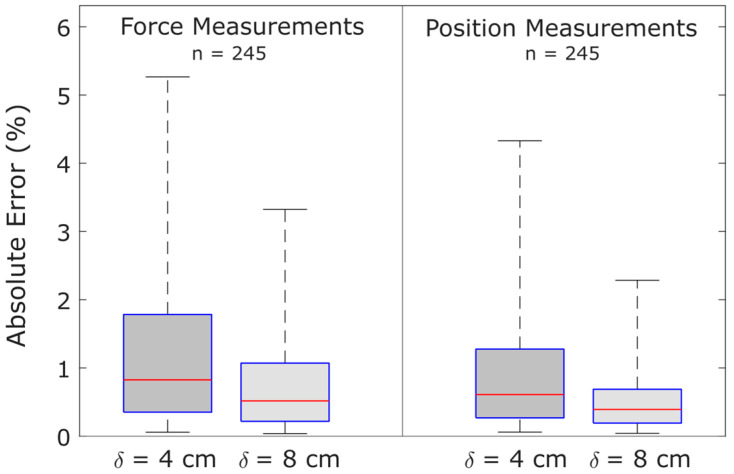
Doubling the distance between strain gauges significantly decreases measurement errors for both force and position. Horizontal box plot lines indicate the 5th, 25th, 50th, 75th, and 95th percentiles of absolute relative error. Data is from Prototype B with applied loads ≥200 g, and all measurement positions described in Section 4.2.4.

**Figure 14 sensors-25-06561-f014:**
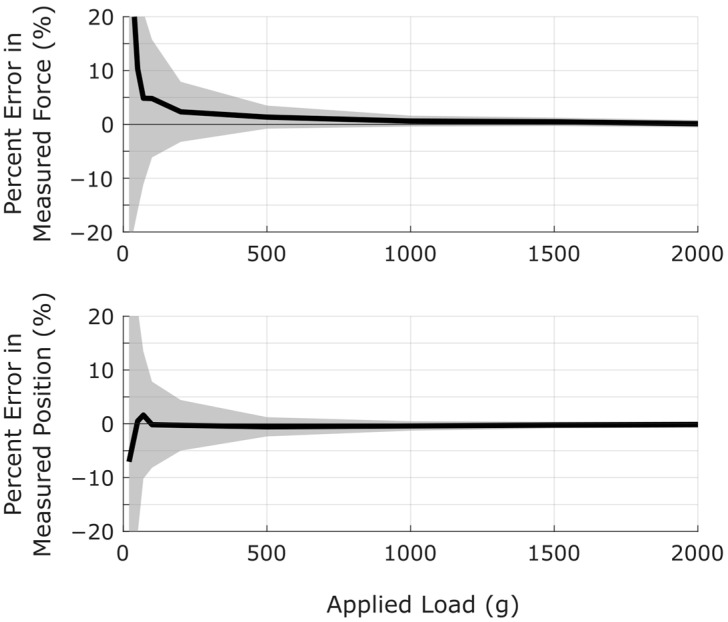
Measurement error decreases as loads increase. Black lines indicate median errors with a corresponding 95% confidence interval shown in gray. Sample sizes were n = 35 at each unique value of applied load.

**Figure 15 sensors-25-06561-f015:**
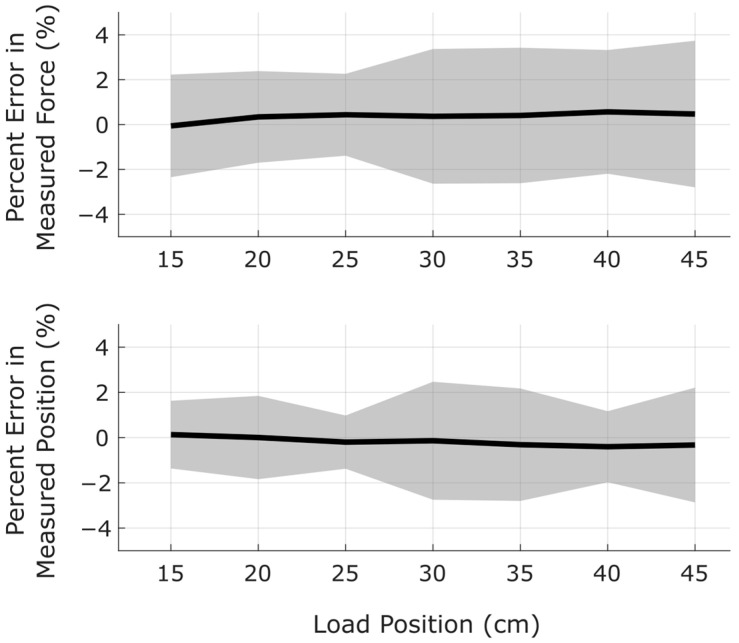
Measurement error exhibits no substantial relationship to the location of the applied load. Black lines indicate median errors. Gray regions represent the 95% confidence interval. Sample sizes were n = 35 at each unique value of position.

**Figure 16 sensors-25-06561-f016:**
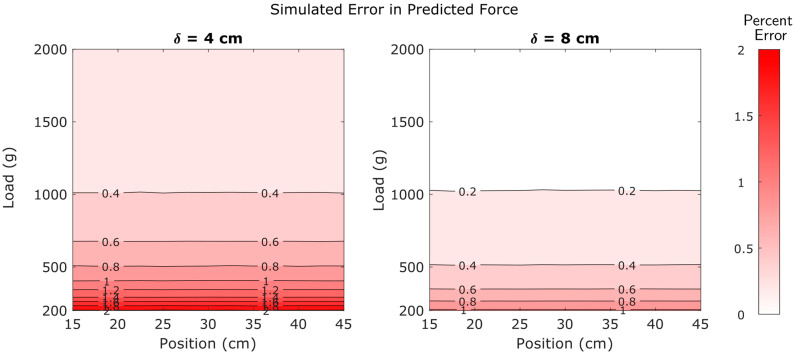
Simulated errors in predicted force values as affected by position, applied load, and strain gauge spacing.

**Figure 17 sensors-25-06561-f017:**
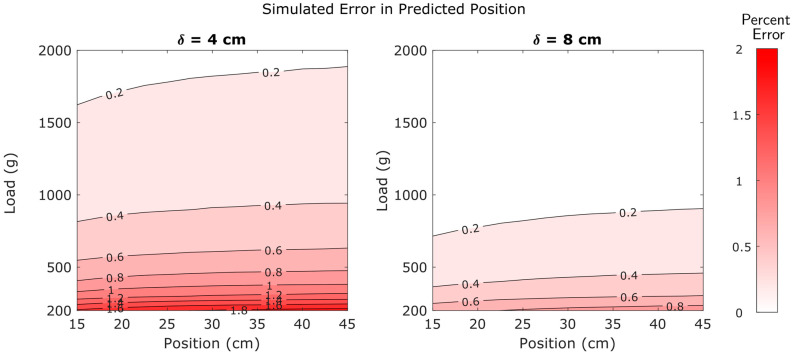
Simulated errors in predicted position values as affected by position, applied load, and strain gauge spacing.

**Table 1 sensors-25-06561-t001:** Physical dimensions of the three sensor prototypes.

Prototype	A	B	C
Beam Profile	Rectangular	Rectangular	Tubular
Material	6061 Alum.	2024 Alum.	6061 Alum.
Length (cm)	80	45	20
Width/Diameter (cm)	5	1.27	1.27
Height (A, B) (cm)	0.318	1.27	n/a
Tube Wall Thickness (C) (cm)	n/a	n/a	0.165
Bridge 1 Position (cm)	2	2	2
Bridge 2 position (cm)	12	6	7
Bridge 3 Position (cm)	n/a	10	n/a

**Table 2 sensors-25-06561-t002:** Masses and positions used for sensor assessments.

Prototype	Masses (g)	Positions (cm)
A	250, 500, 1000, 1500, 2000	15, 20, 25, … 65
B	20, 50, 70, 100, 300, 500, 1500, 2000, 2500, 3000	15, 20, 25, … 45
C	250, 500, 1000, 1500, 2000	10, 12, 14, … 20

**Table 3 sensors-25-06561-t003:** Calibration R^2^ values.

Prototype	A	B	C_x_	C_y_
Bridge Type	Full	Half	Half	Half
Bridge 1 (4 pts)	0.99990	1.00000	0.99997	0.99993
Bridge 1 (9 pts)	0.99942	0.99999	0.99980	0.99966
Bridge 2 (4 pts)	0.99984	1.00000	0.99999	0.99996
Bridge 2 (9 pts)	0.99917	0.99992	0.99976	0.99974

**Table 4 sensors-25-06561-t004:** Percent Error in Measurements.

Prototype	A	B	C
Position	2.110	1.754	0.970
Force	0.643	2.351	1.542
Angle	n/a	n/a	2.618

## Data Availability

The raw data supporting the conclusions of this article will be made available by the authors on request.

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
