# Peer review of "Measurement of Force and Position Using a Cantilever Beam and Multiple Strain Gauges: Sensing Principles and Design Considerations"

_sensors, 2025, doi:10.3390/s25216561_

Round 1

Reviewer 1 Report

Comments and Suggestions for Authors

This paper presents a sensor providing simultaneous measurements of force and position. There are still some issues need to be addressed before its possible publication.

Comments:

1. The force measurement method based on strain gauges and cantilever beam structures is common in force sensors, which is widely used in commercial force sensors. Authors need to clearly state the novelty of the work. For example, in a related work (https://doi.org/10.1016/j.measurement.2024.115140), a force sensor with new structure and corresponding theory was proposed so that the novelty in structure can be emphasized.

2. The authors mention the proposed sensor is suitable for applications in automation and robotics, which needs to be explained. It is recommended that a schematic is added to describe the application.

3. Only the “straightness error” is given for the force sensor. The error of a force sensor includes straightness, repeatability, hysteresis, creep, and other kinds of error in the Force Sensor Test Specification. It is recommended to test the various errors according to the Test Specification, and then define the sensor's accuracy level.

4. What is the resolution of the force and position measurement?

Reviewer 2 Report

Comments and Suggestions for Authors

A method for combined force and position sensing is addressed based on loading at multiple positions of a cantilever beam, which comprises two strain gauges. Details are given on analytical modelling and experimental characterization including error analysis of 3 different device samples. The paper is appropriate for the Journal but should be improved before publication according to the following points:

  1. In Figure 2 (and 3, 4, 5, 7), the selected cantilever dimensions, strain gauge positions, etc. used for calculation should be given. In Figure 6, the parameter range should be given.
  2. Figures 3, 4, 7 appear to shown curves belonging to identical sets of design parameters. Is it possible to merge them into one figure to make the paper more concise?
  3. In line 182, deformation is given as the upper limit of cantilever operation. How about brittle materials like ceramics or silicon?
  4. In line 189, a beam tip is mentioned which is not shown, e. g. in Fig. 2. Should be clarified.
  5. In lines 193 and 194, strain is directly compared with beam moment, which is not possible. The wording has to be improved.
  6. In line 227, the parameter I is mentioned as cross-section, while in Equation (1) it is defined as the beam’s moment of inertia (, which corresponds to common use). Should be clarified.
  7. In Figure 4, the parameters a and d are identical. Is this a necessary condition?
  8. According to line 248, the variable d_1 has no effect on the force ratio, which contradicts Equation (21).
  9. In Eq. (7), “k_i” is meant instead of “k”? In line 156, “… substituting Equation (2) into Equation (6) …” is expected.
  10. “wheatstone” in line 266 should be capitalized (as done consistently otherwise in the manuscript).
  11. Line 266, 267: Shouldn’t the parameter “c” be equal to zero for a balanced Wheatstone bridge?
  12. The parentheses in Equations (24) and (25) are confusing, since they may indicate functional dependences (as in Equation (26)), which seems not to be the case, here. Should be checked.
  13. In line 307, “f” should be replaced by “F”?
  14. What is meant with “Wall Thickness” in Table 1?
  15. In line 419, spelling of “Assesing” and “Acros” should be checked.
  16. In line 448, Figure 12 is meant?
  17. The simulated errors in Figures 17 and 18, left and right part and scale bar should show percent or a unit?
  18. Line 535: “positions” is meant instead of “predictions”?
  19. In line 553, stronger dependence of predicted force on load is mentioned than on position. The physical reason for this finding should be discussed in more detail.
  20. Quantitative values should be estimated and given, e. g., in a table for the ranges and errors of force and position, which can be measured based on the proposed method using the described prototypes and those which could be manufactured in future (including other materials, fabrication techniques, …).
  21. Position and force are typical parameters of tactile surface topography measurement. Stylus instruments or atomic force microscopes for this purpose can be calibrated by scanning along cantilevers as proposed by, e.g.:
    • Erwin Peiner et al., "Silicon Cantilever Sensor for Micro /Nanoscale Dimension and Force Metrology", Microsyst. Technol. 14 (2007) 441-451; https://doi.org/10.1007/s00542-007-0436-8.
    • Jacobo Esteban Munguía-Cervantes et al.,” Si3N4 Young’s modulus measurement from microcantilever beams using a calibrated stylus profiler”, Superficies y vacío, 30 (2017) 10-13; https://www.scielo.org.mx/scielo.php?pid=S1665-35212017000100010&script=sci_arttext.
    • Joachim Frühauf et al., “Silicon Cantilever for Micro/Nanoforce and Stiffness Calibration”, Sensors 22 (2022); https://doi.org/10.3390/s22166253.
    • Georges Adam et al., “An overview of microrobotic systems for microforce sensing”, Annual Review of Control, Robotics, and Autonomous Systems, 7:359–83 (2024); https://doi.org/10.1146/annurev-control-090623-115925.

Such approaches using scanning along a cantilever are an alternative to multiple point loading and should be discussed.

Round 2

Reviewer 1 Report

Comments and Suggestions for Authors

The authors need to carefully check the whole manuscript for typos.

Author Response

Thank you for your review. The following typos and improvements to wording have been made:

Line 2: Capitalized "Using"

Line 81: Changed "dimensions" to "dimensional"

Line 100: Added "the" to "describe how the design parameters..."

Line 108: Added "s" to pluralize "forces" and "positions"

Line 139: Replaced "d" with "di"

Line 140: Added "s" to pluralize "gauges"

Line 254: Removed the word "However" for clarity

Line 353: Removed the word "the" for clarity

Line 388: Removed "in", it was a typo

Line 442: Added "the" to "solve for the corresponding load"

Line 533: Changed "sensor" to "sensor's"

Line 538: Removed "fairly" for clarity

Line 571: Replaced "IT" with "It", fixing typo

Line 578: Removed "relatively" for clarity

Line 608: Replaced "Similar to" with "Like" for brevity

Line 645: Removed "The" for clarity; replaced "are" with "is" to fix typo

Reviewer 2 Report

Comments and Suggestions for Authors

My suggestions for improving the paper were taken into account.

Author Response

Thank you for your review!